# A Preliminary Study of Assessing Gaze, Interoception and School Performance among Children with Neurodevelopmental Disorders: The Feasibility of VR Classroom

**DOI:** 10.3390/children9020250

**Published:** 2022-02-13

**Authors:** Ayako Ide-Okochi, Nobutomo Matsunaga, Hiro Sato

**Affiliations:** 1Graduate School of Health Sciences, Kumamoto University, Kumamoto City 862-0976, Japan; 2Graduate School of Science and Technology, Kumamoto University, Kumamoto City 860-0862, Japan; matunaga@cs.kumamoto-u.ac.jp (N.M.); sato@st.cs.kumamoto-u.ac.jp (H.S.)

**Keywords:** gaze, interoception, school performance, autism spectrum disorder (ASD), attention-deficit/hyperactivity disorder (ADHD), virtual reality (VR), sensory modulation disorder, hyper-focus, children, The Multidimensional Assessment of Interoceptive Awareness (MAIA)

## Abstract

Children with developmental disabilities (DDs) have sensory modulation disorders that interrelate school performance. Virtual reality (VR) has demonstrated the potential to become a neuropsychological assessment modality. This study was conducted to explore the feasibility of the VR classroom for assessing their characteristics of gaze, school performance, and interoception. School-aged children were assigned to the DD group or control group. A VR classroom was designed to evaluate their gaze patterns to distracting events. Interoception was assessed using the Heart Rate Perception test and the Multidimensional Assessment of Interoceptive Awareness (MAIA). The DD group had a significantly longer gaze duration on the virtual teacher during 30–45 s of the VR classroom event (*p* < 0.05). The mean score of the quiz and the Heart Rate Perception test showed a significant tendency to be lower than the children of the control group. The DD group scored significantly lower in six of eight subscales of the MAIA. These results showed the potential of VR classroom to evaluate the difference of sensory modulation between school-aged children with DDs and typically developed children. Future research is necessary to investigate the validity of the VR environment used in this study.

## 1. Introduction

Autism Spectrum Disorder (ASD) and Attention-Deficit/Hyperactivity Disorder (ADHD) are neurodevelopmental disorders that significantly affect cognitive and social development. The estimated prevalence of ASD was 18.5 per 1000 (one in 54) children in the United States [1]. The ADHD/HD worldwide-pooled prevalence was 5.29% [2], and 9.4% of U.S. children have been diagnosed with ADHD [3]. Moreover, these disorders often co-occur. The prevalence of ADHD in children with ASD diagnosis is estimated between 37–85% [4]. In Japan, the number of children suspected of having ASD, ADHD, and Learning Disabilities (LD) is 6.5% in 6–15 year-old children [5]. The number of children with ASD and/or ADHD is estimated a significant ratio worldwide. Therefore, it is indispensable to support them early and enhance their well-being.

According to the Norway study, school refusal behavior is statistically higher in children with ASD than typically developing children despite the absence of intellectual disability [6]. Children with ASD and ADHD are more likely to be absent from school due to social issues, high anxiety, and poor academic performance. Regarding poor academic performance, children with ASD and ADHD supposedly have difficulties concentrating their attention in a noisy school environment, impulse control, and auditory functioning [7]. Children with ASD, ADHD, and both disorders have specific executive functioning deficits in planning, topic shifting, strategy selection, and impulsivity monitoring [8]. Moreover, as children with ASD have weak central coherence, they struggle to integrate and see the context and the whole [9].

It is essential to understand sensory characteristics that may lead to academic failure in children with ASD and ADHD. Abnormal sensory processing and modulation underlie abnormal cognitive behaviors, a defining feature of ASD. Among children with ASD diagnosis, it is estimated that over 90% show symptoms of sensory abnormalities [10]. Recently, in the Diagnostic Manual for Mental Disorders-Fifth Edition (DSM-5) [11], hypo- and hyper-sensory reactivity have been included as diagnostic criteria of ASD. Moreover, there a growing knowledge supporting a relationship between ADHD and sensory over-responsivity [12]. Children with ASD and ADHD have various sensory function impairments that lead to academic failure [7,13]. Children with ASD tend to ignore disturbing stimuli of visual information. ADHD and ASD children are impaired in performing auditory-based tasks in the presence of noise [7]. Children with ASD and ADHD have different sensory systems, such as oral and visual processing scores, as compared to typically developed children [14]. Thus, studies assessing sensory processing characteristics in schools for children with ASD and/or ADHD are necessary [13].

In sensory evaluation, gaze search has recently been used to elucidate the characteristics of children and adults with developmental disabilities (DDs). For example, it has been used to detect gaze patterns specific to ASD [15]. The gaze duration of prominent social information, such as eyes in face images without lip movements, was significantly shorter in the ASD male group while watching 2D pictures and illustrations on the PC screen [15]. Moreover, preschoolers with ASD spent less time gazing at the eyes of people’s faces in the classroom images and at the objects pointed to by the teacher than preschoolers with TD [16]. However, these gaze-search studies used static images (pictures of humans). They did not measure the results in an environment with stimuli, such as sound or movement of people or objects. Therefore, while it is said that children and adults with ASD have a unique vision and visual perception [15], their sensory characteristics have not been fully elucidated, indicating the need for a detailed study. In this study, we hypothesized that students with ASD or ADHD would spend less time looking at the teacher or the teacher’s pointing object and more time looking at the stimulus object in a school environment with audiovisual stimuli when compared to typically developed children, resulting in lower comprehension of the teacher’s explanation.

Furthermore, recently, interoception has been attracting attention. Interoception is a sense of any state inside the body. People with ASD and ADHD have difficulties noticing bodily sensations [17]. However, only five studies have measured interoception using heartbeats in children and adults with ASD, and only one of these studies involved children [18]. In addition to heart rate data, the Multidimensional Assessment of Interoceptive Awareness (MAIA), a scale that measures interoceptive awareness, has been used [19]. However, there is a need to accumulate data on children [20]. This study hypothesized that children with ASD and ADHD would have a lower degree of interoception, as measured by heart rate and MAIA, compared to typically developed children.

Virtual reality (VR) platforms are increasingly being used to assess individuals with DDs [13]. VR environments are of interest in evaluating these sensory and executive function deficits because they provide a realistic experience while allowing for designed experimental stimuli. In a previous study using a VR classroom, adults with ASD and neurologically healthy subjects showed no significant difference in the paper-and-pencil task, the computer task, or the Bimodal Stroop task when there was no distractor. It has been shown that the performance of people with ASD is significantly lower when there is a distractor compared to when there is no distraction [13]. However, previous studies have not thoroughly evaluated atypical sensory processing in children with ASD and ADHD and its impact on school performance using a VR classroom.

The VR classroom environment allows for more accurate measurements than a typical laboratory environment. Because they can be tailored to the preferences of children with DDs and increase cooperation, they are more likely to accurately reflect children’s abilities and highlight the sensory impairments of children with DDs [13]. However, detailed eye measurements are rarely taken in VR classrooms. In addition, the interoception of children who have experienced VR classrooms has not been assessed. The purpose of this preliminary study is to clarify the characteristics of gaze patterns and interoception of children with DDs when disturbance stimuli occur in the VR classroom to obtain hints for understanding their difficulties in school.

## 2. Materials and Methods

### 2.1. Participants

The participants were seven children with DD diagnosis (DD group) and seven children in the control group with typical development (TD group). The diagnosis of DDs was ASD, ASD and ADHD, and ADHD. All children with DDs were recruited by doctors of the Department of Pediatrics at the University of Kumamoto Hospital, Japan. Two doctors diagnosed the participants based on the criteria of the DSM-V [11]. Participants of the control group were not diagnosed or suspected of DDs.

VR is accepted only for those aged seven years or older, according to the guidelines by the XR Association in Japan [21]. All participants were required to be aged between 10 and 18 years, with normal to corrected vision. Exclusion criteria were epilepsy, intellectual disability, and neurological impairments impacting motor movements. The doctors above confirmed that all participants of DD group had 70 or higher Intelligence Quotient (IQ). A parent affirmed that no children from TD group had intellectual disabilities. All participants were administered the same performance test (quiz). All participants did not have any special training experiences, such as yoga, mindfulness, or tai chi, that are likely related to interoception performance [19].

### 2.2. Data Collection

#### 2.2.1. VR Classroom

We designed an experimental application (video) in which the participant sits in a VR classroom surrounded by desks, teachers, students, windows, and blackboards and created visual and auditory disturbance stimuli (Figure 1). In the VR scenario, the partic- ipant sits in the middle seat in the second row, facing and listening to the teacher at the end of the day. The previous study included audiovisual disturbance stimuli, such as books falling on the floor, students raising their hands, paper airplanes flying across the room, planes passing overhead, and a bell ringing along the time axis [13]. This study’s disturbing stimuli include noises from outside the window on the left side of the screen, such as stone-baked potato sales (Video S1), dogs and cats meowing, a teacher pointing at a bulletin board, a notebook falling from the desk, a chime, a change in the screen saver image, and the movement of the second hand of a clock. The stimuli were selected for content that could occur in reality in Japan. For example, the voice selling stone-roasted sweet potatoes is a typical Japanese custom [22]. The sounds of the stone-baked potato sales occurred at 15–30 s; the crying of the dog and cat at 30–45 s; the pointing by the teacher at 45–60 s; and the falling of the notebook at 60–75 s. Unity [23], a 2D/3D game development environment, was used to create the video. The volume was set to the maximum volume of the head-mounted display (HMD). 

Participants viewed the virtual classroom using an HMD, VIVE Pro Eye [24]. The visual stimuli could be viewed at all 360 around the participant. Participants view the VR movie in a seated position. Before the gaze measurement, calibration was performed. The measure was performed with the participant wearing HMD and sitting on a chair. HMD height was adjusted before the measurement because their sitting height varies depending on the participant. The volume was the maximum volume of HMD. The participant was asked to wear HMD after explaining the scenario and quiz after the viewing. To prevent VR sickness [13], we instructed the participants to move not only their eyes but their face and neck when viewing the VR images. There were no participants with symptoms or complaints of VR sickness. The experiment was conducted in a quiet laboratory at the University of Kumamoto. With the consent of the participants and their parents, photographs and videos of the experiments were taken (Appendix B and Appendix C; Appendix A).

#### 2.2.2. Gaze

The VIVE Pro Eye has a built-in eye tracking function and can acquire gazing point data [24]. As a method of determining the gaze (gazing point) in the VR space, a red circle was plotted for each participant every 0.05 s (Figure 2). The red circles were marked sep arately for the teacher, the primary disturbance stimuli, and other locations. The number of red circles was counted every 15 s for each area. The individual data of gaze time was summed up for each group, and the mean value for each group was calculated. 

#### 2.2.3. Interoception

We performed a heartbeat-tracking task based on Schandry’s [25] method to measure the participant’s ability to perceive the heartbeat accurately. Participants were asked to count their heartbeats for a particular time without external cues. The participant sat in a chair in the laboratory and was instructed not to touch the body. The absolute value of the difference between the reported and actual heartbeats in each of the three intervals was calculated. This value was divided by the real heartbeats to calculate the ratio of the discrepancy in the heartbeats. This value was then divided by the actual heart rate to calculate the ratio of heart rate deviation. We subtracted this value from 1 and then calculated the average value for the entire three sections. This value ranges from 0 to 1, and the closer the value is to 1, the more accurately you can feel your heartbeat. In addition, a questionnaire is used to measure the perception of the degree of interoceptive sensation, which can be applied to children over 7 years old [19]. MAIA, which has been developed in Japanese [26], was used. MAIA is 32-item multidimensional instrument that assesses eight concepts: Noticing; Not Distracting; Not Worrying; Attention Regulation; Emotional Awareness; Self-Regulation; Body Listening; and Trusting [19].

#### 2.2.4. Performance Test (Quiz)

To evaluate the effect of disturbing stimuli appealing to the audiovisual sense on academic work, we created a quiz to assess the comprehension of the teacher’s explanation in the VR classroom. In the scenario, the teacher explained the contents of homework, the schedule of tomorrow’s sports practice. Then, the VR teacher made announcements from two student staff groups. Lastly, the teacher alerted her students of an infectious disease. The self-administered quiz consisted of five questions from the contents of the teacher’s description in a 90-s VR movie. The participants wrote their answers in Japanese. At the end of the day, the teacher’s resolutions were atypical and varied according to the schedule of events, etc., making it difficult for children with DDs to grasp the whole picture and respond to the topic. The examinations were: Q1: number of homework assignments, Q2: time of the next day’s school event, Q3: next week’s duty, Q4: the next student staff member mentioned after the student animal keeper, Q5: diseases prevalent in the school.

### 2.3. Data Analysis

We conducted descriptive statistics on attributes (age, sex), gazing time per 15 s, the quiz score, interoception, and MAIA scales for the two groups. After Shapiro–Wilk test was used to confirm the normality of data, the independent samples *t*-test or Mann–Whitney U test was conducted. We set 0.05 as the statistically significant level in the people. All statistical analyses were conducted using IBM SPSS statistics software [27], version 28.

## 3. Results

The results of the Shapiro–Wilk test showed that normality was confirmed regarding age; gaze duration of 0–15 s (notice, teacher), 15–30 s (notice, teacher, other), 30–45 s (teacher), 45–60 s gaze time (notice, teacher, other), and 60–75 s gaze time (notice, teacher, other); interoception score; and MAIA subscale Not Distracting, Not Worrying, Attention Regulation, and Self-Regulation. Therefore, *t*-tests were conducted on these variables. We also conducted the Man–Whitney U test on the 0–15 s gaze time (other) and 30–45 s gaze time (notice, other), and the MAIA subscales of Noticing, Emotional Awareness, Body Listening, and Trusting. For sex, Fisher’s exact test was conducted.

### 3.1. Demographic Characteristics of Participants

There were no significant differences in gender ratio or mean age between the two groups (Table 1).

### 3.2. Group Differences of Gaze

There was a significant difference between the two groups in gazing time to the “teacher” between 30 and 45 s (*p* = 0.025). The gaze time to “others” tended to be significantly lower in the DD group than in the TD group, between 30 and 45 s (*p* = 0.054) (Table 2).

### 3.3. Group Differences of Quiz

The mean score of the quiz showed a significant tendency for the DD group to be lower than the TD group (*p* = 0.065) (Table 3).

### 3.4. Group Differences of Interoception

Interoception and the mean score of the Heart Rate Perception test showed a significant tendency for the DD group to be significantly lower than the TD group (Table 4). The concepts of Noticing (*p* = 0.029), Attention Regulation (*p* = 0.043), Emotional Awareness (*p* = 0.043), Self-Regulation (*p* = 0.015), Listening (*p* = 0.006), and Trusting (*p* = 0.014) of the MAIA scale showed significant differences between the two groups.

## 4. Discussion

There are only a few studies on sensory processing and perception in children with DD [13,18,20]. In addition, few detailed studies have been conducted on gaze data, interoception, and school performance in the VR classroom environment. In this preliminary study, we applied the VR classroom to school-age children and collected measurement data on sensation.

It is widely known that compared to age-matched typically developing individuals, children and adults with ASD spend less time gazing at the eye and face regions [15]. However, there are few data on where students with ASD and ADHD gaze in classrooms with high levels of audiovisual distraction. Students with high-functioning ASD without intellectual disabilities may struggle academically despite having high cognitive abilities associated with academic achievement [28]. ADHD is negatively related to various academic performance indicators, such as GPA [29]. Therefore, the present study provided meaningful results measuring gaze and interoception in an environment similar to a real school.

### 4.1. Gaze

In the VR classroom environment, there was a significant difference between the two groups in the amount of time spent gazing at the “teacher” at 30–45 s. At 30–45 s, auditory disturbance stimuli were generated from the left side of the screen while the “teacher” was explaining. In contrast to our hypothesis, participants with ASD and ADHD continued to gaze at the VR teacher more than their typically developed counterparts, even in situations where they might have been upset by auditory stimuli. This behavioral difference may be related to the influence of higher-level cognition on atypical visual perception in people with ASD [30] or the weak central integration of children with ASD [9]. Moreover, it is known that people with ADHD and ASD have difficulty switching their attention because they concentrate too much on what they are interested in. This hyper-focus phenomenon causes them to lose sight of surrounding sounds and people [31]. It is possible that due to overconcentration, the children with DDs in this study were not as distracted by auditory stimuli as the control group and continued to stare at the teacher.

In school life, children are required to multitask by listening to the teacher and writing notes while their classmates talk and hearing the sounds of stationery and teaching materials [7]. In this study, the gaze pattern of DD group seemed less distracted by auditory stimuli, while they tended to answer the quiz incorrectly. Therefore, it can be inferred that the hypothesis that the participants with DDs more poorly understand the teacher’s explanation due to atypicality of gaze is supported. In other words, it is thought that children with ASD have difficulty in understanding social situations and performing collaborative attention and thus are unable to understand the teacher’s instructions and take appropriate action [16]. Similar to previous findings [13,32], it is possible that the DD group children had impaired executive functions due to being more affected by disturbing stimuli than the TD group. In the future, it will be a challenge to conduct evaluations that approach the academic difficulties experienced by children with DDs more realistically. Teachers need to consider the possibility that a child with ASD or ADHD may gaze at them but understand them differently than intended. VR classrooms have the potential to be useful for assessing and training joint attention between teachers and students. To increase the feasibility of VR classrooms, we propose to develop teaching materials using inexpensive commercial illustrations and HMDs, as in this study.

Second, there may be an effect that children with DDs prefer VR classrooms and animated images; ASD children are more interested in new information technology (IT) technologies and are more willing to participate in measurements using VR classrooms [13]. Children with DDs can struggle to gaze at socially vital information [15]. Still, since the images in this study were created with animation and did not have many facial expressions, so it is possible that the participants in the DD group were not distressed to gaze at them but instead felt that they were desirable visual stimuli. In the future, it will be essential to modify the images that contain more vital social information so that the visual stimuli can be more accurately reproduced, and the effects on the senses of the DD children can be confirmed.

### 4.2. Quiz

Children diagnosed with ADHD may have difficulty solving math problems and reading comprehension problems due to the behavioral characteristics and weak working memory of ADHD [33]. Children with ASD also have difficulty understanding the whole, preferring details because of their weakness of central coherence [9]. These characteristics and functional weaknesses may have contributed to the fact that the group of children with DDs in this study tended to score lower on the quiz than the group of TD children.

### 4.3. Interoception

As measured by the heartbeat tracking task, the interoception score was far from 1 in the group of children with DDs compared to close to 1 in the TD group. Among the subscales of the MAIA, the mean values of Noticing, Attention Regulation, Emotional Awareness, Self-regulation, Listening, and Trusting were significantly lower in the group of children with DDs. The hypothesis that children with ASD or ADHD have a lower interoception than typically developed children was supported. Autistic children showed significantly reduced interoceptive accuracy on the heartbeat-tracking task than typical children [34], replicating adults’ study results [35]. This study is likely to replicate results seen in Palser [34] and Garfinkel [35]. Moreover, according to Garfinkel [35], the awareness subscale of Porge’s Body Perception Questionnaire revealed group differences in interoceptive sensibility between ASD adults and the typically developed. This study’s result regarding MAIA scales is considered in line with Garfinkel’s interoceptive sensibility.

### 4.4. Limitations of This Study and Future Work

First, the total number of participants in this study was small. However, in a previous study using a VR classroom, the number of participants with ASD was also not large [13]. This study is a preliminary study using a pioneering method, the VR classroom. Although the number of participants is small due to the COVID-19 crisis, we obtained valuable results. Second, this study was conducted on developmentally disabled children whom a physician had diagnosed. Although it was confirmed that the children did not have intellectual disabilities, a more rigorous evaluation may have been necessary. Since the sensation is related to developmental characteristics and intellectual level, diagnostic instruments and observation could be used for a more accurate assessment. Third, the VR images in this study were short. In previous studies, a Stroop test of about 20 min was conducted in a VR classroom environment [13]. It is possible that the present study captured only some of the sensory characteristics of children with DDs. However, since we could collect data on school-age children using the VR classroom, we could obtain specific findings, despite its limitations.

## 5. Conclusions

The findings of the present study indicate that children with DDs may gaze at objects and perceive body sensations differently in the VR classroom. Moreover, this study’s findings suggest that they may have difficulties understanding instructions in the school environment where audiovisual stimuli are around. Significantly, our findings also have some implications for future research. Due to the considerably longer gaze time at the same object and the trend of lower comprehension among them, it is hypothesized that future research on hyper-focus and weak central coherence could be helpful. Furthermore, their lower perception of body sensation suggests that evaluation is necessary regarding interoceptive awareness and sensibility among children with DDs. Finally, the usefulness of the VR classroom is suggested as a way to approach these sensory and cognitive atypicalities.

## Figures and Tables

**Figure 1 children-09-00250-f001:**
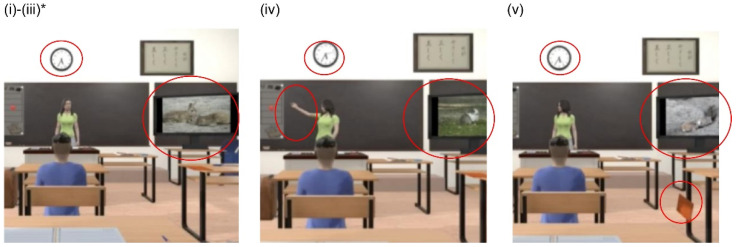
The VR classroom. (**i**) Images at 0–15 s, disturbing stimuli are screensaver and clock, (**ii**) Images at 15–30 s, disturbing stimuli are screensaver, clock, and sound of stone-baked potato sales, (**iii**) Image at 30–45 s, disturbances are screensaver, clock, and crying dog and cat, (**iv**) Images at 45–60 s, disruptors are screensaver, clock, and teacher’s pointing, (**v**) Images at 60–75 s, disruptors are screensaver, clock, and falling note.

**Figure 2 children-09-00250-f002:**
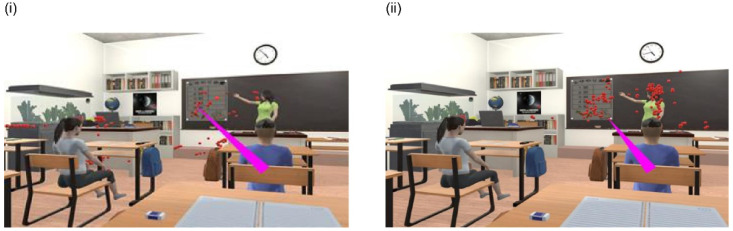
An example of detecting gaze in VR classroom. (**i**) Gazing point at 45–60 s (a child of the group with developmental disabilities), (**ii**) Gazing point at 45–60 s (a child of the control group).

**Table 1 children-09-00250-t001:** Demographic characteristics of participants.

Variables		DD (*n* = 8)		TD (*n* = 7)			
		Frequency	(%)	Frequency	(%)		*p*
Sex	Boy	6	75.0%	5	71.4%		1.000 ^a^
	Girl	2	25.0%	2	28.6%		
Age		Mean	SD	Mean	SD	*t* value	*p*
		12.71	1.60	13.57	0.53	−1.68	0.117 ^b^

DD: developmental disabilities, TD: typically developed, a: Fisher’s exact test, b: *t*–test.

**Table 2 children-09-00250-t002:** Group differences of gaze.

Variables		DD (*n* = 8)				TD (*n* = 7)							
		Minimum Value	Maximum Value	Mean	SD	Minimum Value	Maximum Value	Mean	SD	*t* Value	U	*p*	
Gazing time from 0–15 s	Notice	0.00	2.70	0.98	1.13	0.25	2.55	1.57	0.89	–1.11		0.288	
	Teacher	5.50	15.00	10.83	3.33	3.20	11.60	9.05	2.72	1.12		0.283	
	Others	0.00	6.80	3.19	2.46	1.30	11.55	4.38	3.35		29.50	0.867	^a^
Gazing time from 15–30 s	Notice	0.00	2.95	1.23	1.31	0.00	3.90	1.68	1.35	–0.65		0.527	
	Teacher	3.75	14.85	10.03	3.96	0.00	10.30	7.39	3.45	1.37		0.195	
	Others	0.15	11.25	3.74	3.86	2.05	15.00	5.94	4.19	–1.06		0.311	
Gazing time from 30–45 s	Notice	0.00	3.10	0.87	1.13	0.15	2.05	1.37	0.70		39.50	0.189	^a^
	Teacher	7.80	14.90	11.63	2.36	2.00	11.10	8.21	2.87	2.53		0.025	^*^
	Others	0.00	6.60	2.51	2.37	2.10	12.85	5.42	3.47		45.00	0.054	^† a^
Gazing time from 45–60 s	Notice	1.20	5.55	2.71	1.43	0.00	5.35	3.15	1.77	–0.54		0.600	
	Teacher	0.65	12.40	8.19	4.12	1.70	10.60	7.86	3.21	0.17		0.867	
	Others	0.00	13.15	4.10	4.43	0.60	9.65	3.99	3.21	0.06		0.956	
Gazing time from 60–75 s	Notebook	0.00	2.75	1.11	1.21	0.00	2.60	1.13	0.89	–0.04		0.969	
	Teacher	4.00	11.35	8.90	2.47	3.40	11.55	8.38	2.80	0.38		0.708	
	Others	0.90	8.55	4.99	2.48	2.95	10.00	5.49	2.44	–0.39		0.702	

*: *p <* 0.05, †: *p* < 0.10, No mark: *t*-test, a: Mann–Whitney.

**Table 3 children-09-00250-t003:** Group differences of quiz.

Variables	DD (*n* = 8)				TD (*n* = 7)						
	Minimum Value	Maximum Value	Mean	SD	Minimum Value	Maximum Value	Mean	SD	*t* Value	*p*	
Quiz score	40.00	100.00	70.00	23.90	60.00	100.00	91.43	15.74	−2.02	0.065	^†^

†: *p* < 0.10, *t*-test.

**Table 4 children-09-00250-t004:** Group differences of interoception.

Variables	DD (*n* = 8)				TD (*n* = 7)							
	Minimum Value	Maximum Value	Mean	SD	Minimum Value	Maximum Value	Mean	SD	*t* Value	U	*p*	
Interoception	0.27	0.92	0.61	0.27	0.62	0.98	0.83	0.12	−2.02		0.071	^†^
MAIA Scales												
Noticing	0.00	2.75	0.81	1.01	1.00	4.50	2.29	1.10		47.00	0.029	^* a^
Not Distracting	1.00	5.00	3.25	1.62	1.00	4.67	2.81	1.26	0.58		0.571	
Not Worrying	0.33	3.33	2.29	1.09	1.67	4.67	3.14	0.96	−1.59		0.135	
Attention Regulation	0.00	3.57	1.27	1.17	1.14	5.00	2.65	1.22	−2.24		0.043	^*^
Emotional Awareness	0.00	5.00	1.45	1.58	2.20	4.60	3.03	0.86		49.00	0.043	^* a^
Self-Regulation	0.00	2.25	0.94	0.81	0.00	4.50	2.83	1.67		48.00	0.021	^* a^
Body Listening	0.00	1.33	0.58	0.64	0.67	4.00	2.19	1.09		51.00	0.006	^* a^
Trusting	0.00	5.00	1.38	1.69	2.33	5.00	3.52	1.05		48.50	0.014	^* a^

*: *p* < 0.05, †: *p* < 0.10, No mark: *t*-test, a: Mann–Whitney U test.

## Data Availability

The data presented in this study are available on request from the corresponding author.

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
