# Peer review of "A Preliminary Study of Assessing Gaze, Interoception and School Performance among Children with Neurodevelopmental Disorders: The Feasibility of VR Classroom"

_children, 2022, doi:10.3390/children9020250_

Round 1

Reviewer 1 Report

The central topic of the research article is of great interest although it has been approached from a very particular context and the results are not discussed with previous studies in this regard. El marco teórico es demasiado amplio considerando el análisis y la discusión.

In addition, it would be interesting to focus and delimit the constructs in a clearer way the since the study is highly specific. I would also provide some suggestions/implications for future teachers (strategies, resources, etc.) useful to work the feasibility of VR classroom

The ideas should be better organised (it is not easy to read at some points).

The methodology and the presentation of the results is sufficient however the conclusions are quite general.

Reviewer 2 Report

This study is very interesting. It uses experimental methods to explore the feasibility of VR classroom in evaluating the characteristics of gaze, school performance and interoception of children with developmental disorders (DDs). However, there are the following problems.

1. There are no real experimental photos in the paper. Have you really done the experiment? It is suggested to add photos and videos for reference.

2. Is the VR scene of the experimental group exactly the same as that of the real world? Some unexpected noise and dynamic effects of the picture are the same as those in the real world? These will affect students' gaze and heart rate. The author is suggested to supplement the pictures and experimental details of VR scene.

Author Response

Reviewer 2

This study is very interesting. It uses experimental methods to explore the feasibility of VR classroom in evaluating the characteristics of gaze, school performance and interoception of children with developmental disorders (DDs). However, there are the following problems.

  1. There are no real experimental photos in the paper. Have you really done the experiment? It is suggested to add photos and videos for reference.

  >We revised our manuscript to add real experimental photos. We added photos (Appendix 1, Appendix 2) (page 4, lines 155-179) and videos (Video S1; Video S2) (Supplementary materials).

  1. Is the VR scene of the experimental group exactly the same as that of the real world? Some unexpected noise and dynamic effects of the picture are the same as those in the real world? These will affect students' gaze and heart rate. The author is suggested to supplement the pictures and experimental details of VR scene.

  >We revised our manuscript to add the explanation saying that the VR scene reflected the real world in Japan (e.g. the voice of stone-baked potato sellers) (page 4, lines 138-140). Moreover, we added reference 21) saying hearing the voice of stone-baked potato sellers is Japanese custom. 

This manuscript has not been published or presented elsewhere in part or in entirety, and it is not under consideration by another journal. All study participants provided informed consent, and the study design was approved by the appropriate ethics review boards. All authors have approved the manuscript and agree with submission to your esteemed journal. There are no conflicts of interest to declare.

We thank the reviewers for their comments regarding our previous draft and for your consideration of our revised manuscript. We look forward to hearing from you soon.

Sincerely,

Round 2

Reviewer 2 Report

The revised manuscript can be published. It is suggested to supplement the explanation of the real photos in detail.